# OpenReview forum: "Investigating Memory in RL with POPGym Arcade"
_ICLR.cc/2026/Conference — Submitted to ICLR 2026_

### Official Review · Reviewer_JUnt · 2025-10-25

**Soundness:** 2
**Presentation:** 2
**Contribution:** 2
**Rating:** 4
**Confidence:** 3

**Summary:**

This work presents a POPGym Arcade benchmark and memory evaluation tools in RL. The empirical studies demonstrate that out-of-distribution scenarios can contaminate memory.

**Strengths:**

- This paper introduces POPGym Arcade, an Atari-inspired benchmark, providing fully and partially observable environments.
- This paper proposes memory analysis tools to isolate and study memory capabilities and understand what memory models learn.

**Weaknesses:**

- The overall readability of the paper is poor, and the motivation is not clearly articulated.
- Only one algorithm (PQN) is used to evaluate the proposed benchmark. It would be convincing to include more memory-focused methods, such as R2I [1],
- The current tasks are relatively simple, lacking more challenging ones that could more thoroughly evaluate the model's memory capabilities, such as MemoryMaze [2].
- As shown in Figure 5, the added noise is somewhat too large. It would be better to show results for OOD scenarios with less severe noise to provide a more convincing evaluation.

References:

[1] Samsami et al. "Mastering Memory Tasks with World Models", ICLR, 2024.

[2] Pasukonis et al. "Evaluating Long-Term Memory in 3D Mazes", arXiv preprint arXiv:2210.13383, 2022.

**Questions:**

- Have you tried applying memory models such as Mamba [1] or Transformer [2] to the proposed benchmark?
- Have you tried adding background distractions, similar to those in the Distracting Control Suite [3], to assess the robustness of memory-based agents to perturbations?

References:

[1] Dao et al. "Transformers are SSMs: Generalized Models and Efficient Algorithms Through Structured State Space Duality", ICML, 2024.

[2] Ashish et al. "Attention Is All You Need", NIPS, 2017.

[3] Stone et al. "The Distracting Control Suite -- A Challenging Benchmark for Reinforcement Learning from Pixels", arXiv preprint arXiv:2101.02722, 2021.

---

> ### Author Response · Authors · 2025-11-19
>
> Thank you for taking the time to read and review our work. We will address each point below.
> >> The overall readability of the paper is poor
>
> We have merged the experiments and results section together, which reads much more naturally than the previous version.
> >> the motivation is not clearly articulated
>
> Our primary motivation is stated in the Introduction (Section 1), where we first identify a gap, then address it:
> > Deep RL experiments are sensitive to \[confounding factors\]; especially when combined with memory. This sensitivity makes it difficult to isolate the impact of memory on observability, fairly quantify memory performance, or understand failure modes.
> > To address this gap in understanding, we introduce mathematical tools to isolate and study memory and observability, and to interpret how agents use memory.
>
> >> Only one algorithm (PQN) is used to evaluate the proposed benchmark. It would be convincing to include more memory-focused methods, such as R2I [1].
>
> We have added new experiments in Appendix D:
> - PPO analyses, hinting that policy-gradient methods may demonstrate similar sensitivity.
> - A PQN experiment with $\gamma=0$, which mimics learning a reward predictor in R2I/Dreamer based frameworks.
>
> We have updated the text to explain why we chose to study PQN [3], and how these findings may not hold for MBRL methods like R2I [1]:
> > Model-Based RL: Our study primarily uses PQN, a simple model-free $Q$ learning algorithm, to demonstrate our analysis tools. This was a deliberate choice to isolate pathologies in value estimation from confounding factors in more complex agents. Model-based RL methods learn transition and reward functions using different objectives [1,2], and may not susceptible to the shortcomings we observe in model-free RL. Future studies are necessary to understand learning dynamics for model-based RL.
>
> >> The current tasks are relatively simple ... such as MemoryMaze.
>
> While our environments are Atari-inspired (i.e., 2D, pixel-based), they are designed to be challenging in diverse ways, not just in terms of long-term spatial navigation (as in MemoryMaze [4]).
> For example, our suite includes:
> - **Autoencode** (Appendix H.4), which tests $NC^1$ stack operations (pushing and popping), a known challenge for Linear Recurrent Models [5].
> - **CountRecall** (Appendix H.2), which tests the ability to learn, increment, and query latent counters.
> - **Invisible Tetris** (Appendix H.8), which is unsolved by RL, and is challenging even for human Tetris Grand Masters.
>
> In Appendix H, we describe what a memory model must learn to solve each individual task.
> >> As shown in Figure 5, the added noise is somewhat too large ... less severe noise to provide a more convincing evaluation.
>
> This is a good suggestion. We have updated Appendix E to include experiments with different noise strengths. We see similar phenomena even after reducing the noise strength (e.g., Figure 16). We also highlight that Figure 5b demonstrates this behavior without introducing any observation noise.
>
> [1] Samsami et al. "Mastering Memory Tasks with World Models", ICLR, 2024.
> [2] Hafner et al. "Mastering diverse control tasks through world models", Nature, 2025. Publisher: Nature Publishing Group.
> [3] Gallici et al. "Simplifying deep temporal difference learning" ICLR, 2024.
> [4] Pasukonis et al. "Evaluating Long-Term Memory in 3D Mazes", arXiv preprint arXiv:2210.13383, 2022.
> [5] Merrill et al. "The illusion of state in state-space models" ICML, 2024.
> >> Have you tried applying memory models such as Mamba or Transformer to the proposed benchmark?
>
> Several of our tasks have a Reward Memory Length (RML) of $O(n)$ where $n$ can be thousands of timesteps (Table 1). An optimal policy must consider the full episode, but using a Transformer context window of this size exceeds our available GPU memory. We do include a linear-complexity transformer in all our experiments (**F**ast **A**uto-**R**egressive **T**ransformer).
>
> The Linear Recurrent Unit (LRU) [6] in our experiments is an improved variant of the S5 State-Space Model, and almost identical to the architecture used in Mamba. Our goal is not to determine SOTA memory, but rather demonstrate that these phenomena are not restricted to one specific memory model.
>
> [6] Orvieto et al. "Resurrecting Recurrent Neural Networks for Long Sequences", ICML, 2023.
> >> Have you tried adding background distractions ... to perturbations?
>
> DCS [7] measures how visual distractors impact the policy at the **current** timestep. We study how perturbations **propagate through memory into future** timesteps. Hence our Figure 5b experiments that demonstrate similar behavior **without observation noise**. We have added a short reference to [7] in the text.
>
> [7] Stone et al. "The Distracting Control Suite -- A Challenging Benchmark for Reinforcement Learning from Pixels", arXiv preprint arXiv:2101.02722, 2021.
>
> We hope to have addressed your concerns. Please let us know if any remain.

---

> > ### Comment · Reviewer_JUnt · 2025-11-22
> >
> > Thanks for your responses and clarifications. However, the experiments only evaluated PQN and PPO, and for PPO, the evaluations were restricted to LRU saliency.

---

> > > ### Author Response · Authors · 2025-11-25
> > >
> > > To address the concern that our results rely solely on PQN, we are completing a full sweep using PPO (Observability Gap and Recall Density, Figs. 3-4, 360 runs/seed over all tasks). We expect to update the PDF in 5 days.
> > >
> > > With the addition of PPO, our findings will be validated across 720 distinct task/model/algorithm configurations. This demonstrates that our findings are intrinsic to recurrent model-free RL rather than artifacts of a specific algorithm.

---

> > > > ### Comment · Reviewer_JUnt · 2025-11-25
> > > >
> > > > Evaluating the benchmark with only two algorithms is not sufficient, so I will keep the original rating.

---

### Official Review · Reviewer_VrQS · 2025-10-27

**Soundness:** 3
**Presentation:** 2
**Contribution:** 3
**Rating:** 6
**Confidence:** 3

**Summary:**

Authors conduct controlled analysis on the role of memory in RL environments.
They define evaluation tools like the observability gap, the memory bias, and the recall density.
They design POPGym Arcade of 10 base environments of MDP/POMDP twins to conduct controlled experiments to showcase the utility of the framework.

**Strengths:**

Originality and significance: The motivation of this work stands. It bothers me too that existing literature often merely touch over the role of memory with guesswork, so to me this is a really nice attempt to formally diagnose and scrutinize the learning process empirically in RL with strongly controlled environments.

Quality and clarity: The theoretical framing in Sec 4 and the design in Sec 5 are both solid, reasonable and thoughtful.

**Weaknesses:**

The main complaint I have is the algorithm coverage and depth of analysis: evaluating only on PQN (and its variation) seems lacking to comment in general about RL and memory. Could you try include at least one model-based RL algorithm? What about one that requires replay buffer (and what about the role of different sampling strategies?) There are lots of interesting investigations one can perform with your excellent environment setup.

**Questions:**

1. How do you decide which part of MDP is hidden in their twin POMDP? For example, one could hide the cartpole all together, or one could hide only the pole but not the cart. To me, partial observability is a continuous spectrum and I imagine some hidden states are easier/harder to learn from. Or did you have a general protocol when you convert *any* MDP task to their POMDP compartment?
2. I understand 10 environments are not a lot to work with, but do you see consistent patterns when slicing across dimensions like dense vs. sparse rewards wrt. value smearing?

Suggestions on the writing (not weaknesses and not reasons to reject)
1. "POPGym Arcade is Fast" is a nice-to-have (who doesn't want their eval to run fast?), but is tangential to your analytical contribution. I recommend downplaying the performance, and instead use that space for more memory related experiments and analysis.
2. I found Sec 6 (Experiments) and Sec 7 (Results) hard to read, as the research question, the analysis, and the plot are scattered so I had to jump back and forth multiple times (for each RQ). If somehow one could reserve the locality it would be clearer.

---

> ### Author Response · Authors · 2025-11-19
>
> Thank you for taking the time to read our paper, we hope that you enjoyed it. We are happy to hear that you found our work to be a "nice attempt to formally diagnose" the role of memory. We will address each concern below.
> >> The main complaint I have is the algorithm coverage and depth of analysis: evaluating only on PQN (and its variation) seems lacking to comment in general about RL and memory...What about one that requires replay buffer (and what about the role of different sampling strategies?) There are lots of interesting investigations one can perform with your excellent environment setup.
>
> PQN is simply on-policy Q learning, allowing us to remove target networks and replay buffers as confounding factors. It is the closest we can get to pure value approximation with deep neural networks, and should provide the most general findings. Our library does [implement PPO and provide tuned hyperparameters](https://anonymous.4open.science/r/popgym-arcade-BD90/popgym_arcade/train.py). We have added PPO experiments demonstrating similar phenomena to the Appendices (Figures 14, 17), and we are currently rerunning Recall Density experiments for PPO. We are also currently running a Recall Density study using DQN with a replay buffer. We will update the PDF as results become available.
>
> >> Could you try include at least one model-based RL algorithm?
>
> We argue that MBRL is out of scope for this paper. We aim to study recurrent value functions in this work, rather than specific algorithms. That said, we added an experiment training a reward prediction model by learning $Q$ with $\gamma = 0$ (Figure 13). This is equivalent to learning a reward predictor in a Dreamer-style framework. It also allows us to decouple TD learning from memory learning, and see if the issues persist. We have also updated the limitations section:
>
> > Model-Based RL: Our study primarily uses PQN, a simple model-free $Q$ learning algorithm, to demonstrate our analysis tools. This was a deliberate choice to isolate pathologies in value estimation from confounding factors in more complex agents. Model-based RL methods learn transition and reward functions using different objectives, and may not susceptible to the shortcomings we observe in model-free RL. Future studies are necessary to understand learning dynamics for model-based RL.
>
> >> How do you decide which part of MDP is hidden in their twin POMDP ... when you convert any MDP task to their POMDP compartment?
>
> The **POMDP Required Capabilities** paragraphs in Appendix H explain our thought process for each task's conversion from MDP to POMDP. To summarize, we tried to optimize for POMDP diversity when converting MDPs to POMDPs. Additionally, we designed POMDPs such that they would not admit a near-optimal memory-free solution, which we found the case in some prior work such as visual navigation.
>
> >> I understand 10 environments are not a lot to work with, but do you see consistent patterns when slicing across dimensions like dense vs. sparse rewards wrt. value smearing?
>
> We do see some patterns. The smearing is universal, but it manifests in two different ways depending on the reward structure. Appendix B provides a per-task breakdown of recall density, demonstrating these manifestations.
>
> Tasks with dense rewards (e.g., CartPole, NoisyPole, Skittles, MineSweeper) show more diffuse smears. As you can see in the recall density plots (Fig. 4, and 7-10), the credit is spread out across the entire trajectory. The agent seems to incorrectly believe that all parts of its history are somewhat relevant.
>
> Tasks with sparse/delayed rewards (e.g., Navigator, Tetris) tend to smear credit over the beginning of the trajectory (the τ < 0.33 bucket). This means the agent's Q-value at the end of the episode is dependent on the first frames it saw.
>
> >> "POPGym Arcade is Fast" is a nice-to-have ... for more memory related experiments and analysis. \
> I found Sec 6 (Experiments) and Sec 7 (Results) hard to read....If somehow one could reserve the locality it would be clearer.
>
> This is a good suggestion. We moved the "fast" bits to the appendix, giving us the space to properly explain the results and place the figures. We were able to merge the Experiments and Results sections, which now follows a single chronological thread. We have uploaded the PDF with these changes. Any further suggestions are welcome.
>
> We hope to have addressed your concerns. Please let us know if any remain.

---

> > ### Comment · Reviewer_VrQS · 2025-11-26
> >
> > Thank you for the rebuttal. Most of my questions are answered and my concerns are in the progress of being addressed. I think this is a solid diagnostic paper that makes concrete progress towards our understanding of memory in RL and therefore recommend acceptance (increasing my rating from 6 to 8).

---

### Official Review · Reviewer_MNDG · 2025-10-28

**Soundness:** 3
**Presentation:** 3
**Contribution:** 3
**Rating:** 6
**Confidence:** 4

**Summary:**

This paper provides a new benchmark for research in partially observable reinforcement learning tasks. The benchmark contains re-implementation of 10 common environments in jax, making them hardware acceleratable. The paper also provides implementations of various sequence models combined with the ''PQN" algorithm. Finally, the paper curates 4 metrics (memory gap, observation gap, and input sensitivities) to evaluate different facets of memory-based RL agents.

**Strengths:**

The main strengths of the paper are:
1) Simplicity: The benchmarks are simple and easy to reconfigure.
2) Speed: Because these POMDPs are implemented using jax, they allow researchers to quickly iterate on ideas.
3) Implementation: Multiple memory based models have already been implemented, and the code seems to be readable and clean.

**Weaknesses:**

The papers does not have any major weakness as it does a good job in a niche topic - providing fast benchmark for POMDP research.

**Questions:**

The minor weaknesses of the paper are:
1) I don't understand what Figure 3 is trying to portray. Could you modify the plot to better visualize the data or add a detailed caption?
2) Why didn't the authors implemented the transformer architecture?
3) Figure 4 is also pretty unclear. Is it a POMDP or an MDP? If it is an MDP, then what is the point of using a memory model?

---

> ### Author Response · Authors · 2025-11-19
>
> Thank you for your thoughtful review and constructive questions. Below are clarifications addressing your specific concerns:
>
> >> I don't understand what Figure 3 is trying to portray. Could you modify the plot to better visualize the data or add a detailed caption?
> >>
> >> Figure 4 is also pretty unclear. Is it a POMDP or an MDP? If it is an MDP, then what is the point of using a memory model?
>
> A core finding is that memory models trained on MDPs are using states from throughout the episode to estimate $Q$ values. In an MDP, the agent only needs the current state ($s_t$) to make a good decision. All past states are irrelevant. However, Figure 3 shows bright spots (high gradients) on past, irrelevant states ($s_{t-9}$ through $s_{t-1}$), providing evidence for value smearing (C4).
>
> We have updated the text surrounding these two figures, as well as rewritten the captions for both to improve clarity:
>
> > Figure 3: How do trained agents use memory? We plot pixelwise memory gradients for the LRU (top rows) and GRU (bottom rows), denoting how much each pixel contributes to future value estimate $V(s_t)$ through memory. In these MDPs, $V_*(s_{t})$ is independent of $s_{t-k} \dots s_{t-1}$, yet memory incorrectly smears value credit over uninformative past states, even with a residual connection bypassing the memory model. Smeared value attribution suggests value estimators may not generalize to new trajectories.
>
> > We train policies long beyond convergence and perform qualitative pixel-level gradient visualizations on both MDPs (Figure 3) and POMDPs (Appendix C). We highlight the MDP results because MDPs have a known ground-truth credit distribution: the return depends only on the current state/observation.
>
> > Figure 4: How does the past affect future value predictions? We bin trajectories into thirds, and plot the contribution of each bin on the final $Q$ value. We estimate recall density $\mathbb{E}_{\pi, f}[\delta_Q(\mathbf{x}, \tau)]$ for the start, middle, and end segments $\tau \in [0,0.33),[0.33,0.66),[0.66,1.0)$ of a trajectory, aggregating across models and seeds. All density for MDPs should be in $0.66 \leq \tau < 1.0$, given the Markov property. Instead, we see credit diffusely distributed across trajectories for all models and tasks, demonstrating the value smearing pathology.
>
> >> Why didn't the authors implemented the transformer architecture?
>
> We do implement and evaluate the Fast AutoRegressive Transformer (FART) in all of our experiments (explained further in Appendix K.2). This is a memory-efficient transformer variant.
>
> Most of our tasks are $O(n)$ RML (Table 1), which means that optimal policies must consider the entire trajectory. Given that our episodes can be thousands of timesteps long, our GPUs do not have enough memory to load the full context into a classical transformer.
>
> We hope to have addressed your concerns. Please let us know if any remain.

---

### Official Review · Reviewer_y26h · 2025-10-31

**Soundness:** 2
**Presentation:** 3
**Contribution:** 1
**Rating:** 0
**Confidence:** 5

**Summary:**

This paper introduces a new benchmark, POPGym Arcade, and a corresponding set of mathematical tools designed to analyze the role of memory in deep reinforcement learning agents. The core contribution is POPGym Arcade: a collection of 10 hardware-accelerated, Atari-inspired pixel-based environments (e.g., Tetris, Battleship, CartPole). Its key feature is that each environment provides paired MDP and POMDP variants, which share identical underlying dynamics and state/action spaces, thereby enabling controlled studies using the proposed tools.

**Strengths:**

The paper is well-written and clearly structured. The tools are defined mathematically, and the experiments (e.g., Fig. 2) effectively illustrate their utility. The core finding that return-only comparisons are confounded because the Bias and Gap can be on similar scales is clearly articulated and well-supported.

**Weaknesses:**

The contribution of the benchmark itself (POPGym Arcade) is my main source of contention with this work. The justification for this new suite of "toy" environments is thin, especially given the existing landscape of RL benchmarks.

The paper argues that this new benchmark is necessary to utilize the proposed analysis tools. However, the field is already saturated with Atari-style and grid-world environments, a fact acknowledged by the paper's own related work table (Appendix I). Moreover, these existing benchmarks are sufficient; complex memory challenges are not new. The original Arcade Learning Environment (ALE) contains well-established POMDPs like Montezuma's Revenge and Private Eye that require deep, long-term memory. Other benchmarks like DMLab , MemoryGym , and the original POPGym (which this work builds on) were all created to explicitly test memory. It is unclear why an entirely new suite of 10 games was needed to show these effects. The environments created, such as CountRecall or AutoEncode, feel overly simplistic and purpose-built for the metrics. The RL community is actively trying to move beyond 2D, Atari-style tasks toward more complex, 3D, and physics-based challenges. Proposing a new benchmark in this paradigm, even if it is hardware-accelerated, feels like a step backward and of limited long-term utility to the community.

The paper's main defense is the necessity of paired MDP/POMDP twins. While this enables a very clean measurement of the observability gap, it is not the only way to conduct controlled studies. This design constraint leads to the creation of artificial tasks rather than motivating the tools on problems the community already struggles with. A more compelling paper would have demonstrated these tools on existing, difficult benchmarks (e.g., by creating POMDP-variants of existing MDPs, as done in POBAX Tao et al. (2025)).

**Questions:**

1. The primary critique is the necessity of the POPGym Arcade benchmark. Why were existing, challenging benchmarks (e.g., ALE, DMLab, MemoryGym ) insufficient for demonstrating your analysis tools? What critical aspect do these 10 new games provide that could not be achieved by applying observational masking to existing hardware-accelerated MDPs (e.g., from Gymnax or Jumanji )?

2. The experiments on OOD contamination (Figs 5, 6) are interesting. You note that transformers might mitigate this by discarding OOD inputs once they fall out of the context window. This seems like a critical comparison. Why were transformer-based architectures not included in the main comparisons (Fig 2) alongside RNNs (GRU, MinGRU) and SSMs (LRU, FART)?

3. The related work table (Appendix I) claims POPGym Arcade is unique in having both GPU acceleration and true MDP/POMDP twins. However, the table also lists POBAX (Tao et al., 2025) as having both. Could you clarify the key differentiator from POBAX, which also seems to focus on "benchmarking partial observability"?

---

> ### Author Response · Authors · 2025-11-19
>
> We thank the reviewer for highlighting the paper’s clarity, definitions, and the significance of finding C3. Our primary contribution is the methodology (C1) that enables findings C3–C5. Crucially, we argue that validating C3 and identifying the "value smearing" pathology (C4, C5) was only possible due to the specific design properties of POPGym Arcade.
>
> We address the reviewer's questions below.
> >> The contribution of the benchmark itself ... feels like a step backward ... The paper's main defense is the necessity of paired MDP/POMDP twins ... This design constraint leads to the creation of artificial tasks ...
>
> We respectfully disagree that the benchmark is a "step backward". The environments are not "toy" tasks; they are probes designed to isolate memory pathologies.
>
> In fact, many of these tasks (e.g., CountRecall, AutoEncode, MineSweeper) are hardware-accelerated, pixel-based versions of established tasks from the original, popular POPGym benchmark, which was created specifically to test memory. The reviewer's critique that they are "purpose-built" is correct, we purpose built this new version to enable analysis that was previously impossible.  However, they are far from "simplistic" in practice. As our own return tables in Appendix G show (Tables 2-6), these are among the hardest environments in the entire benchmark for agents to solve.
>
> This combination of conceptual "artificiality" and high "difficulty" is precisely their value. In the MDP variant of CountRecall, we know with certainty that all information from $t < n$ is irrelevant to the value $V(s_n)$. The fact that agents fail so badly on these tasks and simultaneously exhibit profound "value smearing" (Fig. 4) is the core of our C4 finding.
>
> >> Q1: The primary critique is the necessity ... MDPs (e.g., from Gymnax or Jumanji)?
>
> Existing benchmarks were insufficient because they lack one or both of the two features critical for our analysis:
>
> Hardware Acceleration (vs. ALE, DMLab, MemoryGym): To compute our metrics (Gap, Bias) and analyze rare pathologies (Smearing), we needed to run thousands of experiments across many seeds and memory models (2000+ runs, Appendix P). This is simply not feasible using CPU-based benchmarks.
>
> A Unified Observation Space (vs. POBAX, Gymnax-masking): The reviewer suggests we could have just masked existing GPU-benchmarks (e.g., Gymnax), similar to POBAX. As we note in our paper, this masking approach is insufficient for our methodology. In a masked benchmark like POBAX, the MDP state $S$ (e.g., a full pixel frame) and the POMDP observation $\Omega$ (e.g., a partial, masked frame) have different spaces.
>
> This prevents a fair comparison. To measure the Observability Gap (Def 4.1) or Memory Bias (Def 4.2), you would be forced to change the network's encoder (e.g., from a CNN to an MLP) to fit the different observation shapes, introducing a confounding variable. Even if the encoder architectures are similar, the number of parameters will often change (another confounder).
>
> POPGym Arcade uniquely solves this. By design, our MDP and our POMDP observation (a pixel frame) are identical. This allows us to use the exact same network for all MDP and POMDP variants, isolating observability as the sole independent variable. This fulfills the requirement we set in our paper:
>
> >Few works utilize a single shared observation and action space over all tasks, which is necessary to fully control for both parameter count and task difficulty.
>
> >> Q2: The experiments on OOD contamination ... and SSMs (LRU, FART)?
>
> A standard Transformer was not a viable apples-to-apples comparison for our benchmark. As shown in Table 1, several of our tasks (e.g., Invisible Tetris, Autoencode) have a Reward Memory Length (RML) of $O(n)$. A standard Transformer would require a context window that grows with the episode length n, which does not fit in GPU memory. We highlight that we do compare against a linear space complexity transformer (Appendix K.2).
>
> >> Q3: The related work table (Appendix I) claims ... POBAX, which also seems to focus on "benchmarking partial observability"?
>
> We are grateful to the reviewer for catching this, and we apologize for the confusion in Appendix J. The checkmark for POBAX is an error on our part; upon further review it should be a tilde (~) according to our own definition.
>
> Its environments do not share a single unified observation space, and not all are pixel-based. This makes it impossible to apply our Observability Gap (Def 4.1) and Recall Density (Def 4.3) analysis uniformly or to fairly compare a single, fixed model architecture.
>
> POPGym Arcade is therefore unique in providing the complete, necessary suite of features for our analysis: GPU-acceleration, true MDP/POMDP twins, and a unified, pixel-based space. This is what enables our fair, controlled studies on observability. We have revised Appendix J to make these distinctions clear.
>
> We hope to have addressed your concerns. Please let us know if any remain.

---

### Author Response · Authors · 2025-12-01
**General Response**

# General Response
We thank all the reviewers and chairs for taking the time to review our paper. Below, we summarize the rebuttal period.

## Strengths

> The paper is well-written and clearly structured...The core finding...is clearly articulated and well-supported (**y26h**).

> The paper does not have any major weakness...for POMDP research (**MNDG**).

> The motivation of this work stands...this is a really nice attempt to formally diagnose and scrutinize the learning process empirically in RL with strongly controlled environments...the theoretical framing in Sec 4 and the design in Sec 5 are both solid, reasonable and thoughtful (**VrQS**) .

> I think this is a solid diagnostic paper...and therefore recommend acceptance (increasing my rating from 6 to 8) (**VrQS**) .

## Common Weaknessess and Corrections
> One algorithm is not enough **(JUnt, VrQS)**

We repeated all studies using two new algorithms (PPO and DQN), and the results look very similar to before (see Appendix D.2 and D.3). We added reward prediction tasks to simulate learning a reward function in model-based RL (Appendix D.1).

> Environments are too easy **(y26h, JUnt)**

We provide detailed difficulty information in (Appendix G and O). Neither human experts nor trained agents can solve these POMDPs (0.30/1.0 and 0.65/1.0 mean returns respectively). Tasks like Invisible Tetris are even unsolved by Tetris Grand Masters.

> Why no transformers **(y26h, MNDG, JUnt)**?

We include linear transformers in all experiments. Many of our tasks require full-episode context length, which can be thousands of timesteps long. Our GPUs do not have enough memory to train transformers with such context lengths. Currently, We are rerunning short context length experiments using transformers.

> Clarity surrounding results and figures 3, 4 **(MNDG, JUnt, VrQS)**.

We have restructured and rewritten the experiments and results section to follow a single chronological thread. We replaced figures within the paper to improve flow, and rewrote captions for figures 3, 4.

## Further Weaknesses, Questions, and Corrections

> Noise too large in noise experiments (**JUnt**).

We added a sweep over noise values and demonstate similar results for all noise values (see Figure 22).

> Mamba model (**JUnt**)?

Clarified that the LRU model used in all our experiments is nearly identical to Mamba.

> Background distractions similar to Distracting Control Suite (**JUnt**)?

Our Figure 5b demonstrate similar experiments, and we have added a reference to DCS paper.

> What about replay buffer (**VrQS**)?

We reran all experiments using the DQN algorithm and a replay buffer (see Appendix D.3).


## List of Changes

- We have added new PPO and DQN baselines to demonstrate the accuracy and consistency of our contributions.
- We set $\lambda$ to 0 in PQN to simulate a model-based mechanism. We conducted saliency map experiments and observed phenomena consistent with the original PQN.
- We added saliency map experiments and recall density plots for PPO, which yielded consistent results.
- We conducted supplementary experiments for PQN with varying noise levels. We found that even a noise level of 0.1 at 10 time steps significantly influences the memory agent's decision-making within an MDP.
- We performed noise injection experiments on PPO and observed phenomena consistent with PQN. Specifically, in an MDP setting, the agent's memory still attends to frames from the distant past, causing significant changes in the policy distribution $\pi(a|s)$.
- We similarly conducted noise injection experiments on a DQN memory agent under MDP conditions. We observed significant variations in the relative $Q$ values $(A)$.
- We merge experiments and results section. We replaced figures within the paper to improve flow, and rewrote captions for figures 3, 4.

**Regarding the other questions, we have provided detailed responses to each reviewer’s comments.** Once again, we extend our sincere gratitude to all the reviewers and chairs.

---

### Meta-Review · Area_Chair_4VGD · 2026-01-03

**Summary:**

I will list the most important comments that the reviewers noted during the review process:
1) The field of the memory intensive RL is already saturated with Atari-style and grid-world environments.
2) Evaluating only on PQN (and its variation)  and PPO after rebbutal seems lacking to comment in general about RL and memory.
3) It would be convincing to include more memory-focused methods, such as R2I, other model-based methods.
3) The current tasks are relatively simple, lacking more challenging ones that could more thoroughly evaluate the model's memory capabilities.
4) It would be better to show results for OOD scenarios with less severe noise to provide a more convincing evaluation.

**Reviewer Concerns:**

The authors correctly addressed only some comments:
1) About noise: authors have updated Appendix E to include experiments with different noise strengths.
2) Also, during the rebuttal phase, the authors improved the readability of the article.

The authors agreed with some of the shortcomings or they remained unaddressed:
1) About the strength of the benchmark: I can agree with the reviewers that, along with the large number of environments designed to study the features of memory in RL, this benchmark includes few methods and does not offer any special environments.
2) The authors added only one more PPO method, but for an experimental article it is difficult to consider this a serious improvement.

**Reviewer Scores:**

1) Reviewer y26h (score 0) gave an overly harsh assessment and could have raised it to 2 or 4.
2) Reviewer MNDG (score 6) would most likely have left his initial score.
3) Reviewer VrQS (score 6) confirmed explicitly that he would raise his initial score.
4) Reviewer JUnt (score 4) explicitly confirmed that he would leave his initial score.

---

### Decision · Program_Chairs · 2026-01-26

Reject